# Data Agent: Learning to Select
# Data via End-to-End Dynamic Optimization

**Suorong Yang** [1 2]  **Fangjian Su** [1]  **Hai Gan** [1]  **Ziqi Ye** [1]  **Jie Li** [1]  **Baile Xu** [1]  **Furao Shen** [1]  **Soujanya Poria** [3]

## Abstract

Dynamic Data selection aims to accelerate training by prioritizing informative samples during online training. However, existing methods typically rely on task-specific handcrafted metrics or static/snapshot-based criteria to estimate sample importance, limiting scalability across learning paradigms and making it difficult to capture the evolving utility of data throughout training. To address this challenge, we propose Data Agent, an end-to-end dynamic data selection framework that formulates data selection as a training-aware sequential decision-making problem. The agent learns a sample-wise selection policy that co-evolves with model optimization, guided by a composite reward that integrates loss-based difficulty and confidence-based uncertainty signals. The reward signals capture complementary objectives of optimization impact and information gain, together with a tuning-free adaptive weighting mechanism that balances these signals over training. Extensive experiments across a wide range of datasets and architectures demonstrate that Data Agent consistently accelerates training while preserving or improving performance, e.g., reducing costs by over 50% on ImageNet-1k and MMLU with lossless performance. Moreover, its dataset-agnostic formulation and modular reward make it plug-and-play across tasks and scenarios, e.g., robustness to noisy datasets, highlighting its potential in real-world scenarios. Code is available at https://github.com/Jackbrocp/Data-Agent.

[1]Nanjing University [2]CNRS@CREATE [3]Nanyang Technological University. Correspondence to: Baile Xu <xubaile@nju.edu.cn>, Soujanya Poria <soujanya.poria@ntu.edu.sg>.

*Proceedings of the 43rd International Conference on Machine Learning*, Seoul, South Korea. PMLR 306, 2026. Copyright 2026 by the author(s).

## 1. Introduction

Deep learning has made significant progress in recent years, with model architectures becoming increasingly deep and complex to achieve state-of-the-art performance (Touvron et al., 2023; Achiam et al., 2023). However, this progress has created a growing demand for ever-larger training datasets, which leads to substantial training costs. Such costs degrade training efficiency and are usually unaffordable for researchers with limited computational resources. More importantly, large-scale datasets often contain redundancies, further increasing the training burden without necessarily improving model performance. To address these and improve data efficiency, data selection methods (Yang et al., 2024; Xia et al., 2023; Yang et al., 2023b; Zhang et al., 2024) aim to identify highly representative subsets of training data, accelerating training without sacrificing performance. These methods can be broadly categorized into static selection (Tan et al., 2024; Xia et al., 2023; Yang et al., 2025b) and dynamic selection (Qin et al., 2023; Hong et al., 2024; Raju et al., 2021). Static selection identifies a fixed subset of data before training begins, whereas dynamic selection adjusts the data during model training, enabling better adaptation of the training data to the evolving learning process.

While achieving promising results, existing methods face two fundamental limitations. First, most approaches rely on task- or architecture-specific handcrafted metrics to estimate sample importance, e.g., clustering-based statistics (Yang et al., 2025b; Xia et al., 2023) or gradient-derived scores (Tan et al., 2024; Zhang et al., 2024). Such criteria are often tailored to image classification, and are difficult to generalize to paradigms with different supervision and optimization structures, such as object detection. As a result, extending these methods to new tasks typically requires substantial task-specific redesign, hindering scalability and applicability (Qin et al., 2023). Second, sample utility is inherently dynamic and evolves throughout training, yet most methods rely on a converged surrogate model or snapshot-based scores for selection (Qin et al., 2023; Sorscher et al., 2022; Maharana et al., 2023). As observed in (Tan et al., 2024), such evaluations potentially favor samples that are difficult or influential in later training stages, and can be effected by transient training fluctuations. Together, these

limitations raise an interesting and pressing question: *Can we design an agent that adaptively selects data on the fly, while scaling across tasks in a plug-and-play manner?*

Our answer is a resounding *Yes*! In this paper, we propose Data Agent, an end-to-end dynamic data selection framework that adaptively selects training data throughout training. Given that both sample utility and model states evolve throughout training, we formulate dynamic data selection as a sequential decision-making problem, where an agent learns a sample-wise selection policy that co-evolves with model optimization. At each training stage, the agent observes the current model state and determines which samples to prioritize. It is guided by a composite signal that combines a loss-based difficulty measure and a confidence-based uncertainty measure. Intuitively, difficult samples often correspond to underrepresented patterns in the data distribution, while uncertain samples highlight regions near the decision boundary. Our theoretical analyses (Prop. 3.1/3.2) further prove that difficulty and uncertainty signals target complementary objectives, respectively prioritizing samples with larger optimization impact and those yielding higher expected information gains. To automatically balance these objectives over training, we introduce a tuning-free, self-adaptive reward weighting mechanism. Early in training, the agent emphasizes difficult samples to accelerate representation learning, while later it gradually shifts focus toward uncertain samples to refine decision boundaries and improve generalization. This adaptive process enables data selection to co-evolve with the model's learning dynamics. Importantly, due to its dataset-agnostic formulation and modular reward design, Data Agent scales effortlessly across various learning paradigms. It can be seamlessly applied to object detection, semantic segmentation, and LLM instruction tuning, enabling lossless training acceleration in a plug-and-play manner. When augmented with cross-modality semantic consistency signals (Yang et al., 2024), it further exhibits robustness to noisy and corrupted data, highlighting its applicability in real-world settings.

Extensive experiments across a wide range of datasets, architectures, and tasks demonstrate that our method consistently accelerates training while preserving or even improving performance with high efficiency. On large-scale ImageNet-1k (Deng et al., 2009), our method reduces training costs by over 50% while improving performance compared to the full dataset. Beyond image classification, our framework generalizes across tasks such as object detection, semantic segmentation, and LLM instruction tuning. This also highlights its strong cross-architecture generalization, including ResNet (He et al., 2016), ViT (Dosovitskiy et al., 2020), YOLO (Varghese & M., 2024), UperNet (Xiao et al., 2018), and LLaMA (Touvron et al., 2023). Notably, on MMLU (Hendrycks et al., 2021b), our method outperforms the full-dataset baseline by 2% with only 50% of the data on

widely used LLaMA-7B. Moreover, even in noisy datasets, our method demonstrates strong robustness, outperforming existing baselines by at least 8%, underscoring its reliability. Our main contributions are as follows: **1)** We formulate data selection as a training-aware sequential decision-making problem and propose **Data Agent**, an end-to-end framework that learns a sample-wise selection policy co-evolving with model training. **2)** We introduce a composite reward integrating sample difficulty and model uncertainty, together with an adaptive reward weighting mechanism, enabling tuning-free optimization. **3)** With a dataset-agnostic formulation and modular reward structure, Data Agent scales across tasks, architectures, and scenarios, serving as a plug-and-play module. **4)** Experiments demonstrate that our method consistently outperforms SOTA approaches, achieving lossless training acceleration with over 50% reduction in training cost and saving tens to over one hundred GPU hours across deep models.

## 2. Related Work

Data-efficient learning can be broadly categorized into static data selection (Yang et al., 2023b; Paul et al., 2021; Yang et al., 2024; Wang et al., 2026b), dynamic data selection (Qin et al., 2023; Hong et al., 2024; Raju et al., 2021; Yang et al., 2025a;c), dataset distillation (Lei & Tao, 2023; Du et al., 2023; Cazenavette et al., 2025; Li et al., 2025; Zhang et al., 2023; Su et al., 2024), and dataset condensation (Liu et al., 2023; Yang et al., 2023a; Shao et al., 2024; Malakshan et al., 2025). Static pruning is efficient and typically one-shot, but cannot adapt to training dynamics. Dynamic pruning captures evolving sample utility during training, at the cost of modest online overhead. Following dynamic data selection, we propose Data Agent capable of identifying effective training coresets during online training.

### 2.1. Static Data Selection

Static data selection aims to identify a compact yet representative subset of the full datasets before training begins. Models trained on such coresets can achieve results comparable to those on the original dataset. Existing methods are typically based on predefined or heuristic metrics. These metrics can be broadly categorized into importance-criteria-based (Toneva et al., 2018), dataset-distribution-based (Zheng et al., 2023), and optimization-based (Yang et al., 2024). Among importance-criteria-based methods, EL2N and GraNd (Paul et al., 2021) calculate the gradient norm and error-$\ell_2$-norm. Forgetting (Toneva et al., 2018) estimates the samples' misclassification frequency for selection. MoSo (Tan et al., 2024) estimates the effect of removing each sample from the training set, and Memorization (Feldman & Zhang, 2020) assesses the impact of a sample's absence on the model's ability. DUAL (Cho et al.,

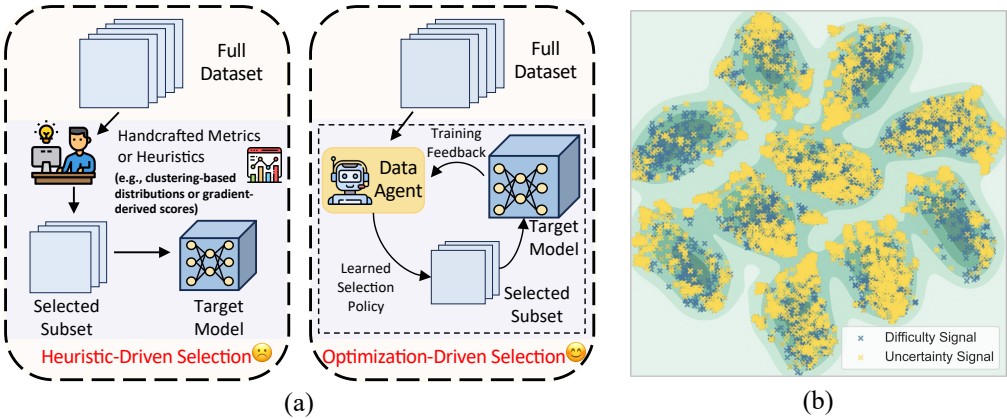

*Figure 1.* (a) **End-to-end dynamic data selection.** Existing methods often rely on handcrafted, task-specific static heuristics to estimate sample importance, limiting the scalability across learning paradigms. In contrast, our framework formulates data selection as a learning problem and jointly optimizes it with model training in a plug-and-play manner, forming a closed-loop, training-aware selection process. (b) **Illustration of data points prioritized by difficulty and uncertainty signals.** The uncertainty signal concentrates on the inter-cluster boundaries and transitional regions, while the difficulty signal focuses more on the sparse cluster areas.

2025) leverages difficulty and uncertainty scores to identify important samples from the early training stage with high efficiency. The work (Yang et al., 2024) leverages multi-modal features to filter out noisy samples by prioritizing semantically aligned samples.

Based on dataset distribution, Herding (Welling, 2009) selects samples closer to the corresponding class centers. D2 (Maharana et al., 2023) estimates sample difficulty based on its neighbors' difficulty, and Moderate (Xia et al., 2023) selects samples with closer distances to the median score. Moreover, CCS (Zheng et al., 2023) evaluates the dataset coverage by extending the classical set cover problem to the distribution cover problem. RL-Selector (Yang et al., 2025b) estimates sample coverage throughout the entire training process and leverages RL to select a fixed subset, requiring the entire training process for each selection ratio. The work (Ramalingam et al., 2023) proposes to compute subsets based on the k-center and uncertainty sampling.

Methods based on optimization algorithms optimize the selected datasets based on gradient matching (Mirzasoleiman et al., 2020b; Killamsetty et al., 2021a), self-supervised metrics (Sorscher et al., 2022), influence functions (Yang et al., 2023b), bi-level optimization (Killamsetty et al., 2021b), facility location function (Mirzasoleiman et al., 2020a; Yang et al., 2023c), temporal dual-depth scoring (Zhang et al., 2024), prediction uncertainty (He et al., 2024), and submodularity function (Nohyun et al., 2023; Kothawade et al., 2022; Wei et al., 2015; Iyer et al., 2021). Despite the promising results, these works face limitations: 1) The predefined or handcrafted metrics hardly work well across architectures and datasets (Qin et al., 2023); 2) To evaluate the training effect of samples, these methods typically introduce substantial additional costs.

## 2.2. Dynamic Data Selection

Dynamic data selection selects data points during online training, allowing the training data to adapt to the models' training stages. The work (Hong et al., 2024) proposes UCB and $\epsilon$-greedy algorithms to select samples with the highest uncertainty during training. The work (Liu & Mirzasoleiman, 2023) proposes a data-efficient framework that extracts small subsets of training data for augmentation to achieve comparable performance to the full datasets. SAS (Joshi & Mirzasoleiman, 2023) observes that the most influential samples for contrastive learning contribute the least to supervised learning, and proposes an algorithm to select subsets that maximize augmentation similarity to the full data. The work (Raju et al., 2021) emphasizes the importance of samples that are used for training a few times and proposes a scoring mechanism based on UCB and $\epsilon$-greedy. Differently, OPUS (Wang et al., 2026a) proposes an optimizer-induced dynamic selection, which formulates data utility through optimizer-induced update dynamics, and GREATS (Wang et al., 2024) optimizes batch quality via Taylor expansion to reduce validation loss. Recently, InfoBatch (Qin et al., 2023) proposes an unbiased dataset pruning method that can accelerate training by pruning less informative samples based on the loss distribution, reducing training costs without degrading performance.

## 3. The Proposed Method

**Overview of the Data Agent.** Dynamic data selection is inherently a sequential decision-making problem that evolves alongside model training, where the utility of data samples changes as the model learns. Here, we propose the Data Agent, a lightweight PPO-based framework that adaptively determines the training data distribution throughout train-

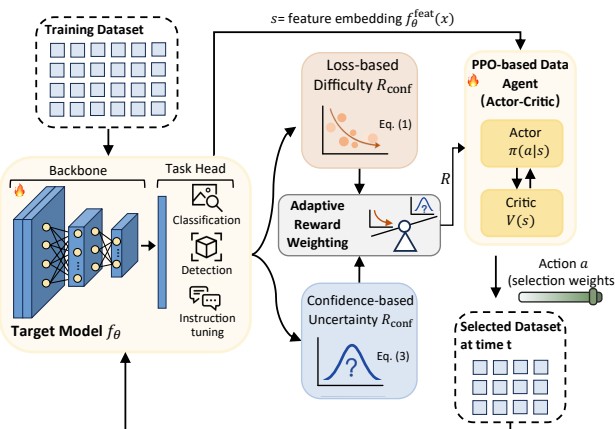

*Figure 2.* The framework of the proposed Data Agent. At each training stage, the agent observes the model state and derives reward signals from standard forward passes. These signals are combined using an adaptive weighting mechanism to guide a PPO-based actor-critic agent, which learns the selection policy. The selected data is used in subsequent training, forming a closed-loop training pipeline where data selection evolves alongside model optimization. Notably, the modular reward design enables the framework to be easily adapted to various learning paradigms.

ing. At each training stage, the data agent observes the current target model state and optimizes a sample-wise policy, guided by two complementary, training-aware signals: a loss-based difficulty signal and a confidence-based uncertainty signal. These signals are directly derived from the forward passes of the target model, capturing the immediate contribution of a sample to empirical risk minimization and the model's predictive uncertainty, respectively. To enable an end-to-end optimization, we introduce an adaptive reward weighting mechanism that automatically adjusts the relative importance of these signals throughout training. This allows the data agent to focus on difficult signals early on to accelerate representation learning, while shifting attention toward uncertainty as training progresses to refine decision boundaries and improve generalization. Notably, due to the dataset-agnostic framework and modular reward structure, our method is highly scalable across diverse tasks, e.g., object detection, and scenarios, e.g., noisy datasets.

### 3.1. Reinforcement Learning Formulation of Data Selection

**Preliminary.** We formulate dynamic data selection problem as a Markov decision process (MDP) (Zhong et al., 2024), denoted by the tuple $\mathcal{M} = (\mathcal{S}, \mathcal{A}, \mathcal{P}, r, \rho, H)$, where $\mathcal{S}$ is the state space, $\mathcal{A}$ the action space, $\mathcal{P}$ the transition kernel, $r$ the reward function, $\rho$ the initial state distribution, and $H$ the horizon length. A policy $\pi(a|s)$ defines a distribution over actions conditioned on the current state $s$. At each step, the agent observes the current state $s$ induced by the target model, updates the policy, and receives rewards derived from the training-aware signals. The agent's goal is to opti-

mize a policy that identifies the most informative training samples for the model.

**State Space.** The state space is defined by the internal representations of the target model, which capture the current training state for each sample. Let $f_\theta(\cdot)$ denote the backbone network parameterized by $\theta$. For a given sample $x$, the state is defined as: $s = f_\theta^{\text{feat}}(x)$, where $f_\theta^{\text{feat}}(\cdot)$ is the feature embedding output from the backbone network. For example, in image classification tasks, this corresponds to the outputs before the final fully connected layer. As training progresses, both the feature space and the model's representation structure change continuously. Thus, the state not only encodes sample-specific information but also captures the model's progress, enabling the data agent to condition its selection policy on both the inherent properties of each sample and the evolving training dynamics.

**Action Space.** The action space is designed for adaptive control over the data distribution. Rather than selecting or discarding samples via discrete decisions, the agent outputs a continuous-valued action for each sample, which avoids the combinatorial complexity and non-differentiability inherent to subset selection. Given a state $s$, the policy $\pi(a|s)$ produces an action $a \in [0, 1]$, representing the selection weight for each sample. This continuous formulation transforms the dynamic data selection problem into a differentiable control problem, which allows for stable and efficient policy optimization in a non-stationary training environment.

**Training-aware Reward Design.** The reward function guides the agent toward adaptive data selection and is computed from training-time forward passes, without relying on a validation set. Importantly, sample utility evolves with model learning: samples that are informative at early training stages may become redundant as the model's representations improve. Thus, we propose a training-aware composite reward that models the interplay between data and the evolving model via two complementary signals: sample difficulty and model uncertainty. This combination encourages focusing on samples that are crucial for optimization and rich in information at the current training stage.

*Loss-based Difficulty Reward:* We introduce a loss-based difficulty reward to estimate the immediate learning difficulty of each sample. Specifically, given the per-sample training loss $\mathcal{L}$ and target model $f_\theta$, the difficulty reward is defined as:

$$R_{\text{diff}}(x_i, y_i) = \mathcal{L}(f_\theta(x_i), y_i). \tag{1}$$

This reward has three key advantages (Qin et al., 2023; Cilimkovic, 2015): 1) loss values can be directly obtained from standard forward passes, avoiding additional computational overhead; 2) it naturally adapts to the evolving training stage of $f_\theta$; and 3) it is task- and architecture-agnostic, making it broadly applicable to various learning paradigms.

**Proposition 3.1.** *Let $f_\theta$ be a network trained with the CE loss $\ell(x, y)$ and softmax outputs $p_\theta(y|x)$. Under a first-order SGD update, the expected magnitude of the parameter update induced by a sample $(x_i, y_i)$ satisfies*

$$\|\nabla_\theta \ell(x_i, y_i)\| \propto 1 - p_\theta(y_i \mid x_i), \qquad (2)$$

*and is therefore a monotonic function of the training loss $\ell(x_i, y_i) = -\log p_\theta(y_i \mid x_i)$.*

Proposition 3.1, proved in Appendix B, shows that prioritizing samples with higher loss accelerates empirical risk minimization by emphasizing those samples with larger optimization impact.

*Confidence-based Uncertainty Reward:* While the difficulty reward captures optimization pressure, it is limited to the likelihood of the annotated class and neglects the uncertainty of the model's prediction. For instance, a loss-based strategy may overlook samples that are correctly classified but still uncertain, particularly those near decision boundaries. These samples are critical for refining the decision boundaries. To address this, we introduce a confidence-based uncertainty reward, derived from the predictive entropy:

$$R_{\text{conf}} = -\sum_{c=1}^{C} p_\theta (y = c \mid x_i) \log p_\theta (y = c \mid x_i), \quad (3)$$

where $C$ is the number of classes. This reward can be obtained via standard forward passes without additional computation cost. By favoring samples with high predictive uncertainty, the data agent is encouraged to focus on boundary-sensitive samples, which are essential for improving decision reliability and generalization.

**Proposition 3.2.** *Let $p_\theta(y|x)$ denote the predicted distribution. Under a first-order SGD update on the CE loss, the expected information gain of learning a sample $x_i$ is proportional to the predictive entropy:*

$$\mathbb{E}_{y \sim p_\theta(y|x)} \left[ D_{\text{KL}} \left( p_{\theta'}(\cdot \mid x) \| p_\theta(\cdot \mid x) \right) \right] \propto H \left[ p_\theta(y \mid x) \right], \qquad (4)$$

*where $\theta' = \theta - \eta \nabla_\theta \ell(x, y)$,*

where $D_{\text{KL}}$ and $H$ are KL-divergence and predictive entropy, respectively. Proposition 3.2, proved in Appendix C, shows that prioritizing uncertain samples approximately maximizes the information gain for model learning. Taken together, Proposition 3.1 and 3.2 highlight the complementary nature of the difficulty and uncertainty rewards, which capture different but crucial aspects of sample utility.

**Adaptive Reward Weighting.** The relative importance of difficulty and uncertainty signals varies across different training stages. In early stages, when model representations are still forming, the difficulty signal drives representation learning by prioritizing challenging samples. As training

progresses, the uncertainty in the model's predictions becomes more informative, enabling finer decision refinement. To capture this dynamic shift and avoid manually tuning, we propose an adaptive reward weighting mechanism that automatically adjusts the contributions of the difficulty and uncertainty rewards based on training dynamics. Since the variance of each reward signal reflects its informativeness (Zhang et al., 2024; Chen et al., 2018), we compute the weighting coefficient $r$ as follows:

$$r = \frac{Var(R_{\text{diff}})}{Var(R_{\text{diff}}) + Var(R_{\text{conf}}) + \epsilon}, \qquad (5)$$

where $\epsilon$ is a small constant for numerical stability. The final reward is computed as:

$$R = r \cdot R_{\text{diff}} + (1 - r) \cdot R_{\text{conf}}. \qquad (6)$$

This formulation allows the agent to adjust its selection focus in an end-to-end, data-driven manner without the need for external hyperparameter tuning. It encourages the agent to balance reducing empirical risk and maximizing epistemic information gain, ultimately enhancing model generalization. Notably, the reward design is modular and extensible, allowing additional task-specific or scenario-specific signals to be adjusted or incorporated as needed, further enhancing the framework's flexibility.

**PPO-based Agent Optimization.** We optimize a sample-wise selection policy using Proximal Policy Optimization (PPO), which dynamically controls the training data distribution during model training. Specifically, the actor and critic networks are parameterized as three-layer MLPs, denoted as $\theta_\pi$ and $\theta_v$, respectively. The policy determines the sample-wise selection weights. Samples with the top-$k$ highest action weights are selected. As the target model and reward signals evolve during training, unconstrained updates to the policy can lead to abrupt changes in the data selection process, potentially destabilizing the joint optimization of the model and the data agent. To mitigate this, we adopt PPO to constrain policy updates and stabilize the co-evolutionary process. PPO's clipped objective enables stable, incremental updates to the policy, while retaining the advantages of policy gradient methods:

$$\begin{aligned} \mathcal{L}_{\text{actor}}(\theta_\pi) = &\mathbb{E}_t[\min(\omega_t(\theta_\pi)\hat{A}_t, \\ &\text{clip}\left(\omega_t(\theta_\pi), 1 - \epsilon, 1 + \epsilon\right) \hat{A}_t)], \end{aligned} \qquad (7)$$

where $\omega_t(\theta_\pi)$ is the probability ratio between the current and previous policies, $\hat{A}_t$ is the advantage estimate at time $t$, and $\epsilon$ controls the range of clipping. In addition, we compute the advantage function using generalized advantage estimation (GAE) (Schulman et al., 2018), which balances bias and variance in non-stationary environments. The temporal-difference residual $\delta_t$ at time $t$ is

$$\delta_t = r_t + \gamma V(s_{t+1}) - V(s_t), \qquad (8)$$

*Table 1.* Comparison with state-of-the-art baselines. All methods are trained using ResNet-18 on CIFAR-10/100 and ResNet-50 on Tiny-ImageNet. Random* refers to randomly selecting samples in each epoch. Some results are from (Qin et al., 2023).

| Dataset | CIFAR-10 | | | CIFAR-100 | | | Tiny-ImageNet | | |
|---|---|---|---|---|---|---|---|---|---|
| Whole Dataset | 95.6 | | | 78.2 | | | 45.0 | | |
| Selection Ratio (%) | 30 | 50 | 70 | 30 | 50 | 70 | 30 | 50 | 70 |
| **Static Selection Methods** | | | | | | | | | |
| Random | 90.2 | 92.3 | 93.9 | 69.7 | 72.1 | 73.8 | 29.8 | 37.2 | 42.2 |
| EL2N (Paul et al., 2021) | 91.6 | 95.0 | 95.2 | 69.5 | 72.1 | 77.2 | 26.6 | 37.1 | 44.0 |
| GraNd (Paul et al., 2021) | 91.2 | 94.6 | 95.3 | 68.8 | 71.4 | 74.6 | 29.7 | 36.3 | 43.2 |
| Forgetting (Toneva et al., 2018) | 91.7 | 94.1 | 94.7 | 69.9 | 73.1 | 75.3 | 28.7 | 33.0 | 41.2 |
| RL-Selector (Yang et al., 2025b) | 91.8 | - | 95.4 | 71.1 | - | 77.6 | 31.1 | - | 44.5 |
| Herding (Welling, 2009) | 80.1 | 88.0 | 92.2 | 69.6 | 71.8 | 73.1 | 29.4 | 31.6 | 39.8 |
| Moderate (Xia et al., 2023) | 91.5 | 94.1 | 95.2 | 70.2 | 73.4 | 77.3 | 30.6 | 38.2 | 42.8 |
| Glister (Killamsetty et al., 2021b) | 90.9 | 94.0 | 95.2 | 70.4 | 73.2 | 76.6 | 30.1 | 39.5 | 43.9 |
| DP (Yang et al., 2023b) | 90.8 | 93.8 | 94.9 | - | 73.1 | 77.2 | - | - | - |
| Self-sup. prototypes (Sorscher et al., 2022) | 91.0 | 94.0 | 95.2 | 70.0 | 71.7 | 76.8 | 27.7 | 37.9 | 43.4 |
| MoSo (Tan et al., 2024) | 91.1 | 94.2 | 95.3 | 70.9 | 73.6 | 77.5 | 31.2 | 38.5 | 43.4 |
| CLIP-Sel (Yang et al., 2024) | 91.9 | 94.5 | 95.0 | 70.8 | 73.7 | 77.0 | 31.7 | 40.0 | 46.0 |
| **Dynamic Selection Methods** | | | | | | | | | |
| Random* | 93.0 | 94.5 | 94.8 | 74.4 | 75.3 | 77.3 | 41.5 | 42.8 | 43.1 |
| UCB (Hong et al., 2024) | 93.9 | 94.7 | 95.3 | - | 75.3 | 77.3 | - | - | - |
| $\epsilon$-Greedy (Hong et al., 2024) | 94.1 | 94.9 | 95.2 | - | 74.8 | 76.4 | - | - | - |
| InfoBatch (Qin et al., 2023) | 94.7 | 95.1 | 95.6 | 76.5 | 78.1 | 78.2 | 42.2 | 43.2 | 43.8 |
| Ours | **95.0** | **95.3** | **96.0** | **77.6** | **78.9** | **79.5** | **44.9** | **47.0** | **49.4** |

where the advantage estimate $\hat{A}_t$ is computed as:

$$\hat{A}_t = \sum_{l=0}^{T-t-1} (\gamma\lambda)^l \delta_{t+l}, \qquad (9)$$

where $\lambda$ is a parameter that controls the trade-off between bias and variance. The critic network is trained to approximate the state-value function $V(s)$ by minimizing the squared temporal-difference error. Given the advantage $\hat{A}_t$ computed by GAE, the value function loss is then given by

$$\mathcal{L}_{\mathrm{critic}}(\theta_v) = \mathbb{E}_t \left[ \left( V_{\theta_v}(s_t) - \hat{A}_t - V(s_t) \right)^2 \right]. \qquad (10)$$

## 4. Experiment

### 4.1. Experiment Setup

**Datasets and Network Architectures.** We evaluate our framework across diverse benchmarks spanning a wide variety of tasks to demonstrate its generalization and scalability. For image classification, we use both coarse-grained and fine-grained benchmarks, including CIFAR-10/100 (Krizhevsky et al., 2009), Tiny-ImageNet (Chrabaszcz et al., 2017), and ImageNet-1k (Deng et al., 2009). We further extend the evaluation to semantic segmentation on ADE20k (Zhou et al., 2018), object detection on MS-COCO (Lin et al., 2015), and LLM instruction tuning on MMLU (Hendrycks et al., 2021b) and AlpacaEval 2.0 (Dubois et al., 2025). Moreover, we evaluate the generalization of our framework to more challenging scenarios, including ImageNet-O (Hendrycks et al., 2021c), ImageNet-Hard (Taesiri et al., 2024), and ImageNet-R (Hendrycks et al., 2021a). To assess its cross-architecture

generalization, we leverage a range of deep models, including ResNet series (He et al., 2016), and ViT series (Dosovitskiy et al., 2020) for classification, YOLOv8 (Varghese & M., 2024) for detection, UperNet (Xiao et al., 2018) for semantic segmentation, and LLaMA 7B (Touvron et al., 2023) for LLM instruction tuning.

**Comparison with State-of-the-arts.** We compare our method with a wide range of static and dynamic data selection methods, including 1) Random, 2) EL2N (Paul et al., 2021), 3) GraNd (Paul et al., 2021), 4) Forgetting (Toneva et al., 2018), 5) RL-Selector (Yang et al., 2025b), 6) Herding (Welling, 2009), 7) Moderate (Xia et al., 2023), 8) Glister (Killamsetty et al., 2021b), 9) DP (Yang et al., 2023b), 10) MoSo (Tan et al., 2024), 11) CLIP-Selector (Yang et al., 2024), 12) Self-sup. prototypes (Sorscher et al., 2022), 13) Random*, 14) InfoBatch (Qin et al., 2023), 15) $\epsilon$-Greedy (Raju et al., 2021), and 16) UCB (Raju et al., 2021).

### 4.2. Performance Comparison

**Results on CIFAR-10/100 and Tiny-ImageNet.** As shown in Table 1, we compare our proposed method with both static and dynamic data selection methods on CIFAR-10/100 and Tiny-ImageNet. Our method consistently outperforms existing methods by a large margin, achieving significant training acceleration without sacrificing model performance. Specifically, it achieves comparable or better accuracy compared to the full dataset with only 50% of the data on CIFAR-100 and 30% on Tiny-ImageNet. While performance tends to decrease as the selection ratio decreases, our method shows the least drop in accuracy compared to full-dataset training. These results underscore the effectiveness of our adaptive,

*Table 2.* Results on ImageNet-1k with a 60% selection ratio using ResNet-50 on an 8-A100 server. We report wall-clock time (h) and total GPU time (GPU hours). Note that, due to high computational and memory costs (Xia et al., 2023), Glister and CG-Score are not reported. Some results are from (Qin et al., 2023).

| Method | Herding | EL2N | GraNd | Forgetting | RL-Selector | SSP | Moderate | CLIP-Sel | UCB | Infobatch | Ours | Whole Dataset |
|---|---|---|---|---|---|---|---|---|---|---|---|---|
| Acc. (%) | 71.1 | 72.3 | 71.0 | 72.5 | 73.4 | 70.0 | 73.1 | 73.2 | 75.8 | 76.5 | **76.8** | 76.4 |
| Time (h) | 10.5 | 10.5 | 10.5 | 10.5 | 10.5 | 10.5 | 10.5 | 10.5 | 10.5 | 10.5 | 10.5 | 17.5 |
| Overhead (h) | >17.5 | >17.5 | >17.5 | >17.5 | >17.5 | >24.0 | >17.5 | >1.6 | 0.03 | 0.0028 | 0.125 | 0.0 |
| Overall (n*h) | >224.0 | >224.0 | >224.0 | >224.0 | >224.0 | >276.0 | >224.0 | >96.8 | 84.0 | 84.0 | 85.0 | 140.0 |

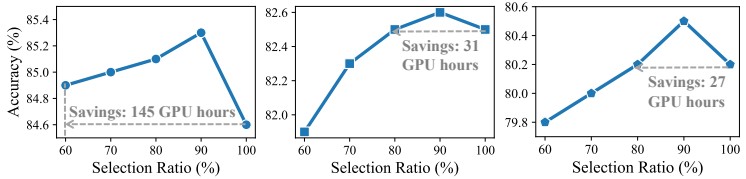

*Figure 3.* Performance and saved costs on ImageNet-1k across Swin-T, ViT-B, and ViT-L on a 4-A100-GPU server. We report the total GPU hours.

*Table 3.* Generalization to LLM instruction tuning using LLaMA-7B with a 50% selection ratio.

| | MMLU | AlpacaEval 2.0 | |
|---|---|---|---|
| | | Win Rate | LC Win Rate |
| Full Dataset | 34.9 | 1.9 | 6.7 |
| Random* | 34.6 | 1.7 | 6.0 |
| Ours | **36.9** | **2.0** | **7.7** |

training-aware selection policy, which dynamically adjusts data selection throughout training to maintain high model performance with fewer data.

**Results on ImageNet-1k.** As shown in Table 2, in addition to the performance comparison, we also compare computation efficiency on ImageNet-1k with a 60% selection ratio. Thus, we report the training time, the introduced overhead, and the total GPU hours for various methods. Notably, our method achieves a 0.4% accuracy improvement over the full-dataset baseline while reducing training costs by nearly 40%, saving over 55 GPU hours. Importantly, since most static methods require training a surrogate model to estimate sample importance, the computational overhead is higher. Thus, our method outperforms static selection methods in both accuracy and efficiency. When compared to other dynamic selection methods, our approach delivers superior performance while maintaining competitive computational efficiency. This efficiency is attributed to our lightweight PPO-based data agent, which operates directly on model features using a compact architecture consisting of just three linear layers, minimizing computation. Meanwhile, the reward can be directly obtained via standard forward passes without introducing complex algorithms. Overall, these results demonstrate that our method delivers an effective and scalable solution on large-scale datasets.

### 4.3. Generalization to More Advanced Architectures

To further evaluate the cross-architecture generalization, we apply the data agent to the training of ViT-based models, including ViT-Base, ViT-Large, and Swin-Transformer. As shown in Figure 3, our method achieves significant training acceleration, with no loss in performance, across different selection ratios. For instance, on ViT-L, our method reduces overall training time by more than 150 GPU hours using only 60% of the data, without sacrificing accuracy. Notably,

*Table 4.* Object detection mAP (%) on MS-COCO using YOLOv8 (Varghese & M., 2024).

| Selection Ratio | 70% | 80% | 90% | 100% |
|---|---|---|---|---|
| Random* | 37.5 | 37.9 | 38.5 | 39.6 |
| Ours | **38.5** | **39.0** | **39.6** | |

*Table 5.* Segmentation mIoU (%) on ADE20K using Uper-Net (Xiao et al., 2018).

| Selection Ratio | 70% | 80% | 90% | 100% |
|---|---|---|---|---|
| Random* | 45.1 | 45.3 | 45.7 | 45.4 |
| Ours | **46.4** | **46.5** | **46.6** | |

at higher selection ratios, our method achieves higher accuracy, outperforming full-dataset training. These results highlight the architecture-agnostic nature of our method, demonstrating its scalability.

### 4.4. Generalization to Different Training Paradigms

**Object Detection.** While most existing data selection approaches cannot be extended to detection tasks, we demonstrate the versatility of our data agent by integrating it into object detection pipelines. Specifically, we apply it to YOLOv8 on MS-COCO, where the sample-wise detection loss naturally serves as the difficulty-based reward signal, without relying on the confidence reward. As shown in Table 4, our data agent achieves lossless or comparable performance when trained with only 70-90% of the data. These results show that our method is not tied to classification-specific supervision or architectures, but can also adapt to dense prediction settings, highlighting its strong scalability and broad applicability.

**Semantic Segmentation.** In addition to object detection, we further assess the generality of our proposed framework by applying it to semantic segmentation, using UperNet on ADE20K. As shown in Table 5, our method not only

*Table 6.* Generalization performance of ResNet-50 trained with our method on ImageNet-Hard/R/O. We report AUPR (%) on ImageNet-O and accuracy (%) on others.

| Selection Ratio | 60% | 70% | 80% | 90% | 100% |
|---|---|---|---|---|---|
| ImageNet-Hard | 14.7 | 15.3 | 15.6 | 15.7 | 14.6 |
| ImageNet-R | 38.0 | 38.2 | 38.5 | 38.7 | 36.2 |
| ImageNet-O | 15.7 | 15.6 | 15.7 | 15.8 | 13.2 |

preserves segmentation performance but also achieves improvements when trained with only 70-90% of the full dataset. These results demonstrate that our method can adapt to dense supervision tasks, highlighting its scalability and architecture-agnostic nature.

**LLM Instruction Tuning.** Beyond vision tasks, our data agent also accelerates LLM instruction fine-tuning, particularly with LLaMA-7B across MMLU and AlpacaEval 2.0. MMLU evaluates multi-domain language understanding, while AlpacaEval 2.0 assesses instruction-following and alignment quality. In this context, the agent dynamically selects informative instruction-response pairs during fine-tuning. As shown in Table 3, even with only 50% of the training data, our method achieves consistent improvements across all metrics. These results validate the effectiveness of our approach for LLM training pipelines, and underscore the framework's modality- and task-agnostic scalability for accelerating large-scale model training.

### 4.5. Generalization under Distribution Shift

To further evaluate the generalization of our method, we conduct experiments on several challenging out-of-distribution benchmarks, i.e., ImageNet-O/R/Hard. These datasets are designed to assess a model's ability to generalize to unseen, more difficult distributions. We compare models trained on the full ImageNet-1k dataset with those trained using our dynamically selected subsets. As shown in Table 6, our method significantly improves performance across all three benchmarks when trained with only 60-90% of the data. These results suggest that our data agent helps the model focus on more informative and representative samples, helping the model learn more generalizable features that are less prone to overfitting specific data distributions. Importantly, the performance improvements come with reduced training costs, enhancing practicality.

### 4.6. Robustness to Noisy Scenarios

Due to the modular design, the proposed data agent can easily incorporate additional signals tailored to specific problem settings. In real-world scenarios, datasets often contain mislabeled or corrupted images (Xia et al., 2023), which can significantly degrade model performance. To address this, we extend the flexibility of our method by integrating a cross-modality semantic alignment signal (Yang et al.,

*Table 7.* Robustness to noisy and corrupted data on Tiny-ImageNet using ResNet-50. The noisy ratio is 20%, and the selection ratios are 20% and 30%. Some results are from (Yang et al., 2024).

| Method | Noisy Dataset | | Corrupted Dataset | |
|---|---|---|---|---|
| | 20% | 30% | 20% | 30% |
| Random | 17.8 | 23.9 | 20.0 | 25.9 |
| Random* | 32.5 | 36.1 | 36.5 | 38.3 |
| CLIP-Sel | 26.1 | 33.1 | 26.1 | 32.1 |
| Ours | **38.1** | **41.9** | **41.9** | **46.7** |

*Table 8.* Effect of the difficulty rewards $R_{\text{diff}}$, uncertainty $R_{\text{conf}}$, and adaptive reward weighting $r$ on T-ImageNet using R-50.

| $R_{\text{diff}}$ | $R_{\text{conf}}$ | $r$ | 30% | 50% | 70% |
|---|---|---|---|---|---|
| | | | 41.5 | 42.8 | 43.1 |
| ✓ | | | 42.1 | 45.0 | 48.2 |
| | ✓ | | 42.2 | 45.8 | 48.4 |
| ✓ | ✓ | | 42.5 | 46.1 | 48.6 |
| ✓ | ✓ | ✓ | **44.9** | **47.0** | **49.4** |

2024) to assess the consistency between visual inputs and textual labels. As shown in Table 7, our method consistently outperforms existing SOTA baselines, achieving over 8% accuracy improvements under noisy label conditions. Notably, such robustness emerges naturally from the unified selection mechanism without any modification to the core learning framework or optimization procedure. These results highlight the adaptability of our method.

### 4.7. Ablation Study

**Effect of Different Components.** As shown in Table 8, we analyze the effect of the difficulty reward $R_{\text{diff}}$, the confidence reward $R_{\text{conf}}$, and the reward weighting $r$. Without using any of these, our method, in the first row, degrades to random selection, which yields the lowest accuracy. This confirms that the proposed reward signals provide a meaningful training-aware supervision signal. Using $R_{\text{diff}}$ or $R_{\text{conf}}$ alone already improves performance, and combining them is consistently better, suggesting that difficulty and uncertainty capture complementary aspects of sample utility. Moreover, using $R_{\text{diff}}$ alone yields slightly lower accuracy, as prioritizing difficulty can lead to the selection of ambiguous or unstable samples. Introducing $r$ further boosts accuracy by dynamically adjusting their relative emphasis over training, effectively creating a self-adjusting curriculum that evolves with the model and improves generalization. Thus, removing any component from the framework degrades performance.

**Effect of Agent Optimization.** To evaluate the effect of the PPO-based agent, we directly use the composite reward in Eq. 6 to select the highest-scoring samples each epoch. As shown in Figure 4, incorporating the PPO agent results in consistent accuracy gains across all selection ratios, validating its ability to capture training dynamics and learn a

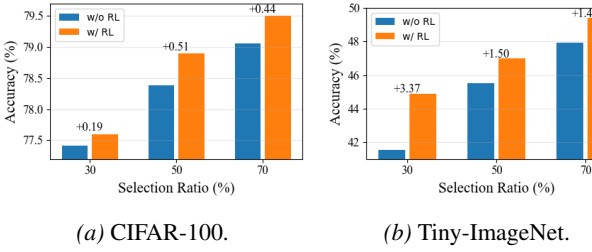

*(a) CIFAR-100.*      *(b) Tiny-ImageNet.*

*Figure 4.* Effect of the RL agent on CIFAR-100 and Tiny-ImageNet under different selection ratios.

coherent, training-aware selection policy.

## 5. Conclusion

We propose Data Agent, an end-to-end dynamic data selection framework that adaptively selects training data throughout training, substantially accelerating training without sacrificing performance. Across diverse datasets, architectures, and scenarios, Data Agent achieves enhanced data-efficient learning, cross-task/architecture generalization, and robustness to noisy or corrupted datasets as a plug-and-play module. We hope this work inspires further research on data-efficient learning and believe Data Agent can serve as a valuable tool for the community, helping reduce computational costs and broaden access to training strong models with limited resources.

## Impact Statement

This paper proposes the Data Agent, a dynamic data selection framework aimed at improving the efficiency of training deep learning models by adaptively prioritizing informative samples. By reducing redundant computation, our approach can lower energy consumption, GPU-hours, and carbon footprint for both academic and industrial training workloads, enabling broader participation by researchers and practitioners with limited resources while also mitigating the environmental footprint associated with large-scale training. Unlike prior task-specific or static selection methods, our dataset-agnostic and plug-and-play design scales seamlessly across learning paradigms and architectures, minimizing engineering overhead when deployed in new domains. More broadly, this work reframes data as an adaptive component that co-evolves with model optimization, opening a data-centric perspective for building efficient, scalable, and robust learning systems.

## Acknowledgement

This work was partially supported by the National Natural Science Foundation of China (Grant Nos. 62495090, 62495094, and 62276127), Fundamental and Interdisciplinary Disciplines Breakthrough Plan of the Ministry of Education of China (No. JYB2025XDXM118), and the "111 Center" (No. B26023). This research/project is supported by the National Research Foundation, Singapore under its AI Singapore Programme (AISG Award No: AISG3-GV2023-010). This work is also supported by the National Research Foundation, Singapore, under its National Large Language Models Funding Initiative (AISG Award No: AISG-NMLP-2024-005), and NTU SUG project #025628-00001: Post-training to Improve Embodied AI Agents. This research is supported by the National Research Foundation, Prime Minister's Office, Singapore under its Campus for Research Excellence and Technological Enterprise (CREATE) programme.

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

# A. The Detailed Algorithm

---

**Algorithm 1** General workflow of our proposed method.

---

**Require:** Training dataset $\mathcal{D}$; batch size $bs$; PPO actor $\pi_\theta$ and critic $v_\theta$; target model $f_\theta$; loss function $\mathcal{L}$; expected selection ratio $r$.

1: **for** each training epoch **do**
2:     Sample a mini-batch $\{(\boldsymbol{x}_i, y_i)\}_{i=1}^{bs}$.
3:     Extract feature representations $f_\theta^{\text{feat}}(\boldsymbol{x}_i)$.
4:     Generate selection actions $\pi_{\theta_\pi}(\cdot \mid f_\theta^{\text{feat}}(\boldsymbol{x}_i))$ and estimate state values $v_i$.
5:     Calculate sample-wise losses $\ell_i = \mathcal{L}(f_\theta(\boldsymbol{x}_i), y_i)$.
6:     Calculate the reward signal $R_{\text{diff}} = \mathcal{L}(f_\theta(x_i), y_i)$ (Eq. (1)).
7:     Calculate the reward signal $R_{\text{conf}} = -\sum_{c=1}^{C} p_\theta(y = c \mid x_i) \log p_\theta(y = c \mid x_i)$, (Eq. (3)).
8:     Calculate the adaptive weights $r = Var(R_{\text{diff}})/Var(R_{\text{diff}}) + Var(R_{\text{conf}}) + \epsilon$ (Eq. (5)).
9:     Calculate total reward $R = r \cdot R_{\text{diff}} + (1 - r) \cdot R_{\text{conf}}$ (Eq. (6)).
10:     Update PPO agent $\pi_\theta$ according to Eq. (7) and (10).
11:     Update target model $f_\theta$.
12:     Construct training subset for next epoch by selecting the top-$k$ samples based on actions $a_i$.
13: **end for**
**Ensure:** trained model $f_\theta$.

---

# B. Proof of Proposition 3.1

*Proof.* Consider a multi-class classification problem, let $(x, y)$ be a training sample, where $y$ denotes the ground-truth class. The cross-entropy loss is given by

$$\ell(f_\theta(x), y) = -\log p_y, \tag{11}$$

where $z = f_\theta(x) \in \mathbb{R}^C$ denotes the logits and $p = softmax(z)$ with $p_y$ being the predicted probability assigned to the true class. For the softmax-CE loss, the gradient w.r.t. the logits has a closed form $\nabla_z \ell(z, y) = p - y$, where $y$ is the one-hot vector corresponding to the true label. Therefore,

$$\|\nabla_z \ell(z, y)\|_2^2 = \|p - y\|_2^2 = (1 - p_y)^2 + \sum_{c \neq y} p_c^2. \tag{12}$$

For the lower bound, note that $\sum_{c \neq y} p_c^2 \geq 0$, we obtain

$$\|p - y\|_2^2 \geq (1 - p_y)^2 \quad \Rightarrow \quad \|p - y\|_2 \geq 1 - p_y. \tag{13}$$

For the upper bound, noting that $\sum_{c \neq y} p_c = 1 - p_y$, the Cauchy–Schwarz inequality implies $\sum_{c \neq y} p_c^2 \leq \left(\sum_{c \neq y} p_c\right)^2 = (1 - p_y)^2$. Substituting into the expression for $\|p - y\|_2^2$ yields

$$\|p - y\|_2^2 \leq 2(1 - p_y)^2. \tag{14}$$

and thus $\|p - y\|_2 \leq \sqrt{2}(1 - p_y)$. Combining both bounds, we obtain

$$1 - p_y \leq \|\nabla_z \ell(z, y)\|_2 \leq \sqrt{2}(1 - p_y) \tag{15}$$

Finally, since the cross-entropy loss $\ell(z, y) = -\log p_y$ is strictly decreasing in $p_y$, larger loss values correspond to smaller $p_y$, which in turn implies larger values of $1 - p_y$. By the above bounds, this leads to larger gradient magnitudes $\|\nabla_z \ell(z, y)\|_2$. $\square$

**Lemma B.1.** *(Connection to Parameter Updates) Let $z = f_\theta(x)$ be the logits produced by a neural network parameterized by $\theta$. By the chain rule,*

$$\nabla_\theta \ell = J_\theta(z)^\top (p - y), \tag{16}$$

*where $J_\theta(z)$ is the Jacobian of logits w.r.t. parameters. Under a standard local smoothness assumption that $\|J_\theta(z)\|_2$ is bounded, the magnitude of the parameter gradient scales with $\|p - y\|_2$ up to a multiplicative factor. Hence, samples with higher loss values typically tend to induce larger parameter updates during stochastic gradient descent.*

## C. Proof of Proposition 3.2

*Proof.* Consider a single SGD update on a sample $(x, y)$ with cross-entropy loss $\ell(x, y) = -\log p_\theta(y|x)$. The updated parameters are:

$$\theta' = \theta - \eta \nabla_\theta \ell(x, y) = \theta + \eta \nabla_\theta \log p_\theta(y \mid x). \tag{17}$$

For simplicity, the change in the predictive distribution can be locally approximated for a small learning rate $\eta$. Using the second-order Taylor expansion, the KL divergence between the predictive distributions before and after the update satisfies

$$D_{\mathrm{KL}}\left(p_{\theta'}(\cdot \mid x) \| p_\theta(\cdot \mid x)\right) \approx \frac{\eta^2}{2} \|\nabla_\theta \log p_\theta(y \mid x)\|^2. \tag{18}$$

Taking expectation over $y \sim p_\theta(y \mid x)$, we have

$$\mathbb{E}_{y \sim p_\theta}\left[\|\nabla_\theta \log p_\theta(y \mid x)\|^2\right] = \mathcal{I}(x), \tag{19}$$

where $\mathcal{I}(x)$ denotes the Fisher information of the predictive distribution at $x$. Thus, the KL divergence scales monotonically with the Fisher information. Since the Fisher information is maximized when the predictive distribution is most uncertain, i.e., $H[p_\theta(y \mid x)]$ is large, the expected KL divergence, and thus the expected information gain, scales monotonically with the predictive entropy. $\square$

Consequently, our confidence-based reward approximates expected Bayesian information gain under SGD dynamics.

## D. Implementation Details

Following experiment settings in (Qin et al., 2023), we train classification models using the OneCycle scheduler with the SGD/LARS optimizer, a momentum of 0.9, a weight decay of 5e-4, and cosine annealing. All images are augmented with commonly used transformations, including random cropping and horizontal flipping. For fairness, we also adopt the annealing and re-scaling techniques introduced in (Qin et al., 2023) across all dynamic data selection methods. Since InfoBatch uses soft pruning with different numbers of selected data points, we report its performance using the same number of forward passes as in our method. For training object detection and semantic segmentation models, we leverage the widely used codebases MMdetection (Chen et al., 2019) and MMSegmentation (Contributors, 2020), respectively, and closely follow the default configuration. Note that some results could not be computed in Table 1 due to the unavailability of open-source code and parameter settings, making it impossible to reproduce. Moreover, our implementation adopts a standard PPO configuration (Schulman et al., 2017) across all experiments: $\gamma = 0.99$, $\lambda = 0.95$ for GAE, and clipping $\epsilon = 0.2$ per update.

## E. Further Analyses on Training Efficiency

*Table 9.* The wall clock time of our method and random baseline using ResNet-50 with a 50% selection on a 4-2080TI server. We report GPU hours.

|  | CIFAR-100 | Tiny-ImageNet |
|---|---|---|
| Random | 3.82 | 5.75 |
| Ours | 3.93 ↑ 2.97% | 5.93 ↑ 3.1% |

To better understand the efficiency of our method, we compare the wall-clock training time of our method against a random selection baseline under identical settings. All experiments are conducted using ResNet-50 with a 50% selection ratio on a 4×2080Ti GPU server. As shown in Table 9, our method introduces only a marginal overhead compared to random selection. This overhead is attributed to the lightweight policy network design and PPO updates, both of which are computationally inexpensive compared to training the target model. Meanwhile, both reward signals are directly derived from standard forward passes, thereby minimizing the additional computational cost. Importantly, this slight increase in wall-clock time is more than compensated for by the substantial reduction in effective training data and the corresponding performance gains reported in previous sections.

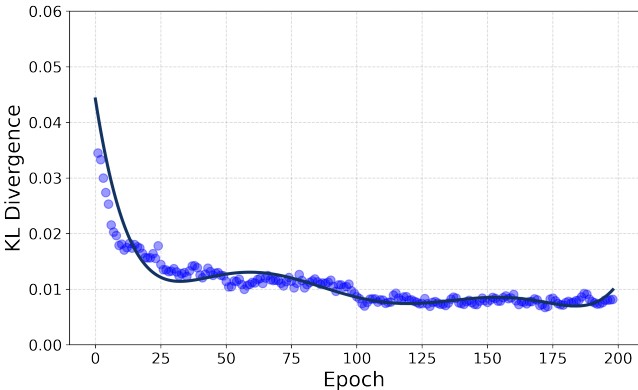

*Figure 5.* KL divergence of the agent-induced selection distribution between consecutive epochs, $D_{\mathrm{KL}}\left(p_t\|p_{t-1}\right)$, on CIFAR-10 with a 20% selection ratio. Higher divergence early in training indicates active adaptation of selected data, while later stabilization suggests a converged yet continuously adjusted selection policy.

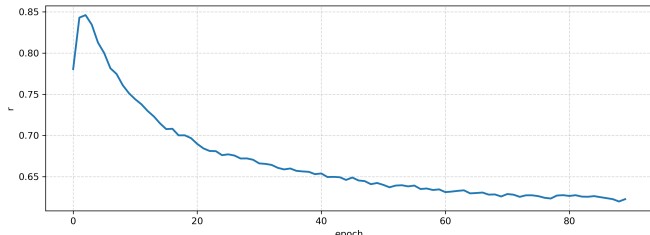

*Figure 6.* The evolving trend of the weighting coefficienct $r$ throughout the training process.

## F. Analysis of the Agent's Evolving Selection Behavior

In this section, we analyze how the agent's selection behavior evolves over training by measuring the divergence of the selected dataset distributions across epochs. As illustrated in Figure 5, we observe a higher divergence in early training epochs, followed by a gradual decrease and stabilization as training progresses. This trend is consistent with the intended behavior of a training-aware selector: the agent actively explores and adjusts the selection policy rapidly when representations are still forming, and divergence decreases later but remains non-zero as the target model approaches convergence. This indicates that the selected subset continues to evolve rather than collapsing to a fixed coreset, aligning with our goal of adaptive, stage-aware data selection.

Moreover, as shown in Fig. 6, $r$ starts high (0.85+) and gradually decreases to around 0.6, indicating a smooth shift from difficulty to increased uncertainty awareness, while the difficulty remains effective.

## G. Discussion and Future Work

In this paper, we propose Data Agent, an end-to-end dynamic data selection framework that adaptively selects training data throughout optimization, achieving training acceleration without sacrificing performance. In this section, we discuss some potential limitations and future work for our method. 1) Our current implementation trains the agent jointly with the target model. While the agent is lightweight and adds only marginal overhead, a promising next step is to pretrain a generic Data Agent that can be rapidly adapted to new datasets, architectures, and tasks (e.g., via few-shot adaptation), further reducing amortized cost and improving practical deployment. 2) While the co-evolving design introduces non-stationarity, it is well-controlled and gradual: the target model is updated through incremental gradient steps, so the state and reward distributions shift smoothly over training. Under such co-evolution, PPO serves as a stable online optimizer, where clipped updates and advantage estimation help mitigate moderate drift. Empirically, this design yields stable training and consistent gains across diverse tasks. 3) Data Agent currently specializes in on-the-fly selection. More broadly, the same training-aware agentic formulation could be extended to compositional data-centric AI, such as curriculum pacing, data scheduling, or data generation.

