# OpenReview forum: "Data Agent: Learning to Select Data via End-to-End Dynamic Optimization"
_ICML.cc/2026/Conference — ICML 2026 regular_

### Official Review · Reviewer_7Gft · 2026-02-20

**Soundness:** 3
**Presentation:** 3
**Significance:** 3
**Originality:** 3
**Overall Recommendation:** 5
**Confidence:** 4

**Summary:**

The authors point out that previous data pruning approaches rely on task- or architecture-specific metrics, and most methods depend on a converged model or snapshot-based scores. To mitigate these issues, the authors propose Data Agent, which adaptively determines the training data distribution by leveraging both loss-based difficulty and confidence-based uncertainty. The proposed method is validated through extensive benchmarks, including basic image classification on CIFAR-10/100, Tiny-ImageNet, and ImageNet. Furthermore, the authors conducted experiments on object detection, semantic segmentation, and LLM instruction tuning tasks, demonstrating the effectiveness of the approach. The results confirm that the method significantly reduces training costs while preserving accuracy to a certain extent; notably, on ImageNet, the accuracy even surpasses full dataset training at specific selection ratios. Finally, the authors verified that Data Agent works well in more realistic scenarios, such as those involving noisy and corrupted data.

**Compliance With Llm Reviewing Policy:**

Affirmed.

**Final Justification:**

The authors’ rebuttal has satisfactorily addressed my main concerns, including those regarding reward scaling, baseline performance, and comparisons with other SOTA methods. Accordingly, I have increased my score to 5 following the rebuttal.

**Key Questions For Authors:**

1. Regarding the reward design, I have concerns about the scale sensitivity of the weighting coefficient $r$, which is determined by the variances of the two reward components. Specifically, $R_{\mathrm{diff}}$ is defined based on the loss values of individual data points. If the loss function $\mathcal{L}$ is scaled by a constant $c$, its variance $Var(R_{\mathrm{diff}})$ would increase by a factor of $c^2$, directly shifting the balance of the weighting coefficient. A similar issue arises with $R_{\mathrm{conf}}$, which is calculated using entropy; its scale and variance are inherently dependent on the number of classes in the dataset. Could the authors clarify how they address these scale-dependency issues? It is unclear whether the current framework requires explicit reward normalization or a manual rescaling process to ensure that one reward component does not numerically dominate the other across different tasks or loss scales.
2. The authors claim that the difficulty signal ($R_{\mathrm{diff}}$) drives representation learning in the early stages, while uncertainty ($R_{\mathrm{conf}}$) helps refine decision boundaries in later stages. However, there is no empirical visualization of the weighting coefficient $r$ over time to support this narrative. Does the value of $r$ actually start close to 1 and gradually decrease as training progresses? I would like to see a plot showing the evolution of $r$ throughout the training process to verify if the agent's behavior aligns with the authors' stated intuition.
3. Similarly, to better understand the training dynamics, I would like to see a qualitative analysis of the samples selected at different stages. Specifically, it would be highly informative to visualize which types of samples are prioritized during the early phase versus the later phase.
4. Most (static) data pruning literature reports performance at high pruning ratios. Does Data Agent maintain effectiveness under extreme computational constraints (e.g. 10% selection ratio)?
5. Based on the reward design, $R_{\mathrm{diff}}$ assigns higher rewards to samples with larger losses. This suggests that label noise or corrupted images would naturally be assigned high rewards and prioritized for selection. However, the results in Table 7 show strong performance on noisy/corrupted datasets, implying that the agent successfully filters these samples. Could the authors provide empirical evidence (e.g., selection frequency for noisy vs. clean samples) to verify if the agent is indeed filtering them? Furthermore, I would appreciate a clarification on the underlying mechanism that prevents the agent from over-selecting these high-loss noisy samples despite the current reward formulation.

**Limitations:**

The authors discuss the limitations in Appendix G.

**Strengths And Weaknesses:**

### **Strengths**

1. The paper is well-organized and presents a compelling motivation. The authors clearly identify the limitations of prior data pruning methods, specifically their lack of applicability across different tasks and architectures, as well as their inability to capture the dynamic nature of sample utility. To address these gaps, the authors propose Data Agent, an innovative approach that learns a sample-wise selection policy guided by both difficulty and uncertainty. The effectiveness of the proposed method is thoroughly validated through a diverse set of experiments, complemented by comprehensive ablation studies that provide deeper insights into the contribution of each component.
2. The proposed method, Data Agent, introduces a novel perspective to the data pruning literature. While most existing data pruning methods rely on specific, pre-defined, or hand-crafted metrics, Data Agent treats the dynamic data selection problem as a MDP. By adopting a RL framework, the authors successfully optimize a policy that learns to identify the most informative training samples for the model.
3. The paper stands out for its extensive experimental evaluation, which goes significantly beyond the standard image classification benchmarks prevalent in prior literature. To demonstrate the broad applicability across various tasks and architectures, the authors conducted experiments not only on image classification but also on object detection (MS-COCO), semantic segmentation (ADE20K), and LLM instruction tuning (MMLU and AlpacaEval). This diverse range of tasks effectively showcases the versatility of the proposed method. Furthermore, the authors verify the robustness and efficacy of Data Agent through experiments on challenging and realistic scenarios, including ImageNet-O/R/Hard as well as noisy and corrupted data on Tiny-ImageNet, where the method consistently demonstrates strong performance.

### **Weaknesses**

1. The current evaluation lacks baseline comparisons for tasks beyond image classification. While I acknowledge that many pruning methods are task-specific, the authors should, at a minimum, provide results for a Random$^*$ selection baseline across all experiments (Object Detection, Segmentation, Instruction Tuning, and Robustness) to justify the necessity of the Data Agent.

    Furthermore, for image classification, the paper would benefit from comparisons with recent SOTA static pruning methods such as CCS [1], Dyn-Unc [2], and DUAL [3]. Although static methods generally underperform compared to dynamic ones, comparing against DUAL [3] would be particularly informative as it claims robustness to noisy/corrupted datasets and outperforms dynamic methods like InfoBatch [4]. Furthermore, since DUAL shares a similar motivation with Data Agent by incorporating both difficulty and uncertainty into its metric, a discussion and direct comparison is necessary to better distinguish the advantages of the proposed RL-based approach.

2. I have identified several inconsistencies in the notations within Section 3.1 that need to be addressed:
    - In Equation (1), $R_{\mathrm{diff}}(x_i)$ should be corrected to $R_{\mathrm{diff}}(x_i, y_i)$ as it is defined using the loss function. Similarly, in Equation (3), $R_{\mathrm{conf}}$ should be explicitly written as $R_{\mathrm{conf}}(x_i)$.
    - In Equation (4), the term $D_{\mathrm{KL}}$ and $H$ is used without a formal definition; it should be explicitly defined at least verbally (presumably as KL-divergence and entropy).
    - In Equation (7), $r_t(\theta)$ is used, which creates a conflict with $r$, already defined as the weighting coefficient in Equation (5). A unique notation should be used for the policy ratio in Eq. (7).
    - The term $\epsilon$ is used in both Equation (5) and Equation (7), likely representing different values; distinct notations should be used to avoid confusion.
    - In Equation (9), the term $\lambda$ is used without an explicit definition.
    - In line 203, $\theta$ is defined as a network parameter. However, the authors use the same $\theta$ for both the actor and critic networks. It is unclear whether they actually share the same parameters. According to the "Results on ImageNet-1k" paragraph, these networks seem to be parameterized as three-layer MLPs. This detail should be moved to the Methodology section, and distinct notations should be assigned to differentiate the parameters of the agent from those of the main training model ($\theta$).
3. The authors should clarify the trade-offs between static and dynamic pruning methods in the Introduction or Related Work. Static pruning focuses on constructing a "coreset" to achieve both storage and training efficiency, whereas dynamic pruning aims to improve training efficiency within a single run where the full dataset remains accessible. Consequently, dynamic methods typically outperform static ones as they can leverage more total information throughout the training process. In a similar vein, the authors should follow the convention of works like InfoBatch [4] by categorizing the methods in their results tables into static and dynamic pruning. This would better highlight the methodological positioning of Data Agent and clarify its performance advantages.
4. The overall presentation of the Algorithm 1 in Appendix A requires revision. Currently, the use of vector notation is inconsistent, appearing only in this section, and much of the process is explained only verbally. For instance, the transition from line 4 to line 5 is abrupt and lacks a clear logical flow. Rather than simply citing equations, the authors should present a self-contained algorithm that integrates all components into a unified framework. I recommend redefining the algorithm to ensure that each step is mathematically explicit and that the notation remains consistent with the rest of the paper.
---
**References**

[1] Coverage-centric Coreset Selection for High Pruning Rates, ICLR 2023.

[2] Large-scale Dataset Pruning with Dynamic Uncertainty, CVPR Workshop 2024.

[3] Lightweight Dataset Pruning without Full Training via Example Difficulty and Prediction Uncertainty, ICML 2025.

[4] InfoBatch: Lossless Training Speed Up by Unbiased Dynamic Data Pruning, ICLR 2024.

---

> ### Author Rebuttal · Authors · 2026-03-31
>
> Dear Reviewer 7Gft,
>
> We sincerely thank you for the meticulous review and valuable suggestions. We appreciate your recognition of our work's unprecedented versatility, novelty, and broad applicability.
>
> We respond below:
> - **Q1: Random\* baseline for non-classification tasks.**
> - **A1:** We added Random* results for all non-classification settings (Tab. D-1/2/3/4). Our method consistently outperforms Random*, further validating the effectiveness of Data Agent.
>
> Since the revised manuscript can not be updated at the current stage, we will include these in Tab. 3/4/5/7 in the final version.
>
> **Table D-1:** Object Detection.  $S_r$ is the selection ratio.
> |$S_r$ (%)|70|80|90|
> |-|-|-|-|
> |Random*|37.5|37.9|38.5|
> |Ours|**38.5**|**39**|**39.6**|
>
> **Table D-2:** Sementic Segmentation.
> |$S_r$ (%)|70|80|90|
> |-|-|-|-|
> |Random*|45.1|45.3|45.7|
> |Ours|**46.4**|**46.5**|**46.6**|
>
> **Table D-3:** Instruction Tuning.
> ||MMLU|Alp 2.0 WR|Alp 2.0 LC|
> |-|-|-|-|
> |Random*|34.6|1.7|6.0|
> |Ours|**36.9**|**2.0**|**7.7**|
>
> **Table D-4:** Robustness.
> |$S_r$ (%)|20|30|20|30|
> |-|-|-|-|-|
> ||Noisy||Corrupted||
> |Random*|32.5|36.1|36.5|38.3|
> |Ours|**38.1**|**41.9**|**41.9**|**46.7**|
> - **Q2: Comparison with more suggested works [a-c].**
> - **A2:** We compared our method with the suggested methods (Table D-5), and further with DUAL under noisy/corrupted settings (Table D-6). We used the reported results of prior methods and the same training settings. The results show that our method consistently achieves higher accuracy. We will include these references in Sec. 2.1 in the final version.
>
> **Table D-5:** Comparison with suggested works.
> |$S_r$ (%)|30|50|70|
> |-|-|-|-|
> |C-10|
> |CCS|92.7|95.1|95.3|
> |Dyn-Unc|91.8|95.4|95.5|
> |DUAL|91.8|95.0|95.3|
> |Ours|**95**|**95.3**|**96**|
> |C-100|
> |CCS|68.9|73.8|76.8|
> |Dyn-Unc|64.3|74.2|77.7|
> |DUAL|66.4|74.6|77.4|
> |Ours|**77.6**|**78.9**|**79.5**|
> |IN1k|
> |CCS|67.8|70.5|72.3|
> |Dyn-Unc|63.5|68.3|70.9|
> |DUAL|68.6|71.5|72.8|
> |Ours|**71.9**|**73.2**|**74.0**|
>
> **Table D-6:** Robustness comparison with DUAL on T-In1k using R34.
> ||Noisy||Corrupted||
> |-|-|-|-|-|
> |$S_r$ (%)|20%|30%|20%|30%|
> |DUAL|37.9|42.8|34.9|40.0|
> |Ours|**42.2**|**44.9**|**46.3**|**48.6**|
>
> [a] CCS; [b] Dyn-Unc; [c] DUAL
> - **Q3: Several notation inconsistencies in 3.1.**
> - **A3:** We corrected **all** the mentioned issues in Sec. 3.1 to ensure consistency.
> - **Q4: Trade-offs between static and dynamic pruning.**
> - **A4:** We will add a brief discussion in Sec. 2, para 1: *"Static pruning is efficient and typically one-shot, but cannot adapt to training dynamics. Dynamic pruning captures evolving sample utility during training, at the cost of modest online overhead."*
> - **Q5: Method categorization in results.**
> - **A5:** We will categorize methods into static and dynamic in the results tables.
> - **Q6: Improvement of Alg. 1.**
> - **A6:** We have revised Alg. 1 in Appendix A to ensure consistent vector notations and more explicit mathematical formulations in the final version.
> - **Q7: Clarification on the reward scale.**
> - **A7:** Both the loss-based and entropy-based rewards are normalized to [0,1], ensuring a consistent scale across tasks. Thus, the adaptive weighting depends on relative dispersion rather than absolute magnitude, mitigating scale sensitivity. We will clarify this in the final version.
> - **Q8: Evolution of the weighting coefficient $r$.**
> - **A8:** We examined the evolution of r during training. Consistent with our intuition, r starts high (~0.85+) and gradually decreases to ~0.6, indicating a smooth shift from difficulty to increased unvertainty awareness, while the difficulty remains effective. We will include this visualization in Appendix.
> - **Q9: Qualitative analysis of selected samples.**
> - **A9:** We will include qualitative visualizations in Appendix in the final version.
> - **Q10: Extreme computational constraints.**
> - **A10:** We further evaluated our method at an extreme selection ratio of 10% (Table D-7). Our method achieves higher accuracy than other baselines, showing that Data Agent remains effective under extreme budgets.
>
> **Table D-7:** Accuracy at 10% selection ratio.
> ||C10|C100|T-IN1k|
> |-|-|-|-|
> |Random*|92.5|72.6|36.2|
> |Dyn-Unc|59.7|34.6|-|
> |CCS|85.9|54.2|-|
> |DUAL|55.0|34.4|-|
> |Ours|**93.3**|**74.1**|**37.6**|
> - **Q11: Robustness under noisy scenarios.**
> - **A11:** As discussed in Sec. 4.6 (lines 415-423), our framework incorporates an additional cross-modality semantic alignment signal into the reward to evaluate the semantic consistency. Although noisy/corrupted samples may have high loss, they typically exhibit low semantic alignment, which suppresses their overall reward and prevents them from being selected.
>
> To verify this, we analyzed the average proportion of noisy samples in selected subsets (Table D-8). The results show that noisy samples are rarely selected.
>
> **Table D-8:** Average noise ratio (%) in the selected subests during training.
> |$S_r$ (%)|20|30|
> |-|-|-|
> |Random|20.8|29.8|
> |Ours|**4.0**|**5.8**|

---

> > ### Author Rebuttal · Reviewer_7Gft · 2026-04-01
> >
> > Thank you for the detailed response. It resolved most of my concerns, including the issue with the reward scale. After reading the rebuttal and other reviewers' comments, I have raised my score to **5: Accept**.
> >
> > I hope to see the rebuttal points added to the revised manuscript. I also strongly suggest sharing the code for reproducibility.

---

> > > ### Author Response · Authors · 2026-04-01
> > >
> > > Dear Reviewer 7Gft,
> > >
> > > We would like to express our sincere gratitude to reviewer 7Gft for acknowledging our work and providing insightful comments. We will incorporate the discussed clarifications into the final manuscript to improve clarity and completeness. We also appreciate your suggestion on reproducibility and will make our code publicly available.
> > >
> > > Thanks again for the time and effort in reviewing our work.
> > >
> > > Authors

---

### Official Review · Reviewer_9oia · 2026-03-01

**Soundness:** 3
**Presentation:** 3
**Significance:** 3
**Originality:** 3
**Overall Recommendation:** 5
**Confidence:** 4

**Summary:**

This study introduces an end-to-end dynamic Data selection framework called Data Agent to accelerate deep learning training by prioritizing the processing of samples with high information volume. This system regards data selection as a sequential decision-making problem and uses reinforcement learning strategies to make the selection algorithm evolve synchronously with the optimization state of the model. The Data Agent combines the difficulty signal based on loss and the uncertainty signal based on confidence, and adopts an automatic weighting mechanism to balance these two goals at different training stages. Experiments have proved that this method can reduce the training cost of fine-tuning instructions for ImageNet-1k and large language models by more than 50% while maintaining or even improving performance. Its design features high universality and plug-and-play characteristics, capable of providing stable acceleration effects across various tasks and architectures such as image classification, object detection, and semantic segmentation.

**Compliance With Llm Reviewing Policy:**

Affirmed.

**Final Justification:**

The author's rebuttal addressed my main concerns

**Key Questions For Authors:**

1) At present, the reward function combines difficulty and uncertainty. In scenarios of extreme category imbalance or long-tail distribution, will these two signals conflict? Is the adaptive weighting mechanism still effective in these extreme cases?

2) The paper mentions that pre-trained general agents are future work. Then, can a Data Agent trained on an architecture (such as ResNet-50) be directly used for data selection in ViT or other tasks without updating the weights?

3) How to choose reinforcement learning methods? This study utilized the PPO algorithm. Will other types of RL algorithms have any impact on the research?

**Limitations:**

yes

**Strengths And Weaknesses:**

**Strengths**
1) Different from the traditional static screening, the selection strategy of the Data Agent co-evolves with the model optimization and is capable of capturing the evolving utility of samples at different training stages

2) This method is dataset-independent and architecture-independent, and experiments have been conducted on multiple tasks

3) The agent itself is very lightweight, and the reward signal is directly obtained from the standard forward propagation, with extremely low additional overhead

**Weaknesses**
1) Although PPO is used to stabilize policy updates, jointly optimizing the target model and reinforcement learning agents is inherently more complex than single supervised learning and may carry the risk of unstable training

2) In the early stage of training, both the target model and the agent are in an unconverged state. At this time, the reliability of the selection weights generated by the agent may be relatively low

3) Current implementations typically require training an agent from scratch for each specific training task, and the "universal selection capability" across tasks has not yet been achieved

---

> ### Author Rebuttal · Authors · 2026-03-31
>
> Dear Reviewer 9oia,
>
> Thank you for the constructive suggestions and insightful comments. We appreciate your recognition of our work's strengths, e.g., high universality, extensive experimental results, and extremely low overhead.
>
> We respond below:
>
> - **Q1: Stability of joint optimization.**
> - **A1:** Thanks for the insightful comment. While joint optimization introduces theoretical complexity, it is essential for capturing evolving learning dynamics and sample utility online, **enabling plug-and-play adaptation without requiring a separately pretrained agent or embedding model**. In practice, our framework remains stable and lightweight: the agent adds negligible overhead (<3%, Tab. 2/9), and the clipped PPO objectives stabilize policy updates.
>
> To further assess stability, we conducted additional analyses: 1) Fig. 5 in Appendix F shows an exploration-to-stabilization pattern, with KL divergence between selection distributions gradually decaying and stabilizing. 2) Compared with directly using the reward as a score (w/o RL, Table C-1) over 10 random seeds, our method achieves both higher accuracy and lower variance.
>
> **Table C-1:** Accuracy on T-IN1k using R50 (Mean ± Std, %).
> |$S_r$ (%)|30|50|70|
> |-|-|-|-|
> |w/o RL|41.5±0.5|45.5±0.5|47.9±0.6|
> |Ours|**44.9±0.3**|**47.0±0.2**|**49.4±0.2**|
> - **Q2: Reliability in early stages.**
> - **A2:** Insightful comment. While both the agent and target model are initially unconverged, our framework is designed to operate online from scratch, without requiring any separate pretraining. The analyses in Fig. 5 in Appendix F show that the KL divergence between subsets is higher early on and then stabilizes, indicating exploration first rather than premature biased selection.
>
> To further address the comment, we conducted additional analyses: 1) We test a variant (R-Ours) that uses random selection in the first 10% epochs (Table C-2). Our method achieves higher or comparable accuracy, showing that the agent captures useful signals early on. 2) Compared with random selection, our method reaches the same test accuracy with **over 30% fewer training steps** in early stages.
>
> **Table C-2:** Effect of warm-up random selection on T-IN1k using R50.
> |$S_r$ (%)|30|50|70|
> |-|-|-|-|
> |R-Ours|43.7|47.0|49.0|
> |Ours|**44.9**|**47.0**|**49.4**|
> - **Q3: Universal selection capability.**
> - **A3:** We would like to clarify that our method is designed to be task-agnostic and plug-and-play at the framework level, requiring no task-specific redesign or separate pretraining, and is empirically validated across diverse tasks and architectures. Training the agent from scratch enables adaptation to different data distributions and optimization dynamics. Since the agent is extremely lightweight (<3% overehad, Tab. 2/9), this online adaptation is practical and efficient. In contrast, most existing data selection methods are tightly coupled with specific tasks through handcrafted, task-dependent metrics, which limits their scalability. Our approach instead provides a unified framework that generalizes across settings.
>
> As discussed in Appendix G, developing such a pretrained universal agent remains future work, which is also a key challenge in data selection research.
> - **Q4: Robustness under long-tail distributions.**
> - **A4:** We further conducted additional experiments on long-tailed Places-LT (Table C-3). The results show that our method remains robust under imbalanced distributions. This is because the adaptive weighting balances difficulty and uncertainty during training: tail samples often exhibit higher loss and uncertainty and thus are prioritized, while redundant head samples are down-weighted. Since both signals are normalized and bounded, neither dominates the reward.
>
> **Table C-3:** Evaluation on Places-LT (closed-set).
> |$S_r$ (%)|80|90|100|
> |-|-|-|-|
> |Ours|34.6|36.1|34.5|
> - **Q5: Cross-arch generalization of data agent.**
> - **A5:** Insightful comment. As suggested, we evaluated by directly applying an agent trained with R-18 to ViT-Small without retraining (Table C-4). The transferred agent achieves nearly identical performance, indicating promising cross-architecture generalization.
>
> **Table C-4:** Cross-arch transfer on C-100 (R-18 -> Vit-small).
> |$S_r$ (%)|30|50|70|
> |-|-|-|-|
> |T-Ours|88.0|89.2|90.2|
> |Ours|87.9|89.2|90.1|
> - **Q6: Choice of RL methods.**
> - **A6:** We use PPO for its stability and low hyperparameter sensitivity. Our framework is not tied to PPO. We further evaluated SAC and DDPG (Table C-5), which achieve similar results across datasets and selection ratios. This suggests that the gains mainly stem from our MDP formulation and the policy learning mechanism.
>
> **Table C-5:** Generalization across RL algorithms.
> ||SAC|DDPG|Ours|
> |-|-|-|-|
> |**C-10**|
> |30|94.9|95.0|95.0|
> |50|95.2|95.4|95.3|
> |70|96.0|95.9|96.0|
> |**C-100**|
> |30|77.7|77.5|77.6|
> |50|78.7|79.0|78.9|
> |70|79.3|79.3|79.5|
> |**T-IN1k**|
> |30|44.8|44.9|44.9|
> |50|46.9|47.1|47.0|
> |70|49.4|49.2|49.4|

---

> > ### Author Rebuttal · Reviewer_9oia · 2026-04-03
> >
> > I appreciate the supplementary experiments you conducted based on my questions, so I have decided to further increase my score.

---

> > > ### Author Response · Authors · 2026-04-03
> > >
> > > Dear Reviewer 9oia,
> > >
> > > We would like to express our sincere gratitude to Reviewer 9oia for acknowledging our work and providing insightful comments. We are glad that our responses have addressed your comments. Thank you again for your time and effort in reviewing our work.
> > >
> > > Authors

---

### Official Review · Reviewer_Pgoo · 2026-03-05

**Soundness:** 2
**Presentation:** 4
**Significance:** 3
**Originality:** 3
**Overall Recommendation:** 3
**Confidence:** 3

**Summary:**

This paper proposes Data Agent, a PPO-based reinforcement learning framework for dynamic data selection during training. The agent observes feature embeddings from the target model, outputs continuous selection weights per sample, and is guided by a composite reward combining loss-based difficulty and entropy-based uncertainty, balanced by a variance-based adaptive weighting mechanism. The authors evaluate across image classification (CIFAR-10/100, Tiny-ImageNet, ImageNet-1k), object detection (MS-COCO), semantic segmentation (ADE20K), and LLM instruction tuning (MMLU, AlpacaEval), claiming over 50% training cost reduction with lossless or improved performance across multiple architectures including ResNet, ViT, YOLO, UperNet, and LLaMA-7B.

The experimental coverage is impressively broad and the results are strong on paper. However, I find the method description fundamentally incomplete — the paper never specifies how continuous selection weights translate into actual data selection, and the MDP formulation has critical gaps that make it difficult to understand what the method is actually doing. Without these details, the contribution cannot be properly evaluated.

**Compliance With Llm Reviewing Policy:**

Affirmed.

**Key Questions For Authors:**

- How exactly are continuous action values converted to discrete sample selection? What mechanism enforces the target selection ratio?
- What are the PPO hyperparameters (γ, λ for GAE, clipping ε, number of PPO epochs per update)? These are never reported.
- How does the agent handle the non-stationarity of the MDP? The target model changes every step, so the reward function and state distribution shift continuously. Standard PPO convergence assumes a stationary MDP.

**Limitations:**

The most critical omission is the selection mechanism itself. The agent outputs continuous weights a ∈ [0,1] per sample, but the paper never explains how these weights become discrete selection decisions. Is there a threshold? Top-k selection? Weighted sampling?

Algorithm 1, step 10 simply says "Select samples for next-epoch training based on the sample scores" — while this is similar to coreset selection, I would like some details on how this is done.

The PPO agent processes each sample independently — π(a|s) takes a single sample's embedding and outputs a scalar weight. There is no inter-sample interaction in the policy. This means the agent is functionally a learned per-sample scoring function, not a sequential decision-maker. It is unclear what PPO's temporal machinery (advantage estimation, clipped updates, value function) contributes beyond a simpler learned scoring approach.

he paper never clarifies whether the agent scores the entire dataset each epoch (requiring a full forward pass just for scoring, undermining efficiency claims) or operates within batches (which precludes global ranking).

**Strengths And Weaknesses:**

- The experimental breadth is genuinely impressive, spanning classification, detection, segmentation, and LLM fine-tuning across five architecture families. Few data selection papers attempt this scope.
- The overhead is minimal (~3% wall-clock increase over random selection), owing to the lightweight 3-layer agent and rewards derived from existing forward passes.
- The noisy data experiments show strong robustness (8%+ over baselines), and the OOD generalization results on ImageNet-O/R/Hard suggest the selected subsets lead to more generalizable features.
- The adaptive reward weighting mechanism is a sensible idea — shifting from difficulty-driven to uncertainty-driven selection as training progresses mirrors curriculum learning intuitions.

---

> ### Author Rebuttal · Authors · 2026-03-31
>
> Dear Reviewer Pgoo,
>
> We sincerely thank you for the careful review and insightful suggestions. We appreciate your recognition of our work's strengths, e.g., impressive experimental breadth, minimal overhead, and strong robustness/generalization.
>
> We provide responses to address the comments below:
>
> - **Q1/2/3: Clarification on the mechanism for translating continuous weights to discrete selection and target ratios enforcement.**
> - **A1/2/3:** Thanks for the question. We would like to clarify that our framework adopts a **deterministic Top-$k$ selection** over the continuous action values. At each epoch, we construct the subset $D_t$ by selecting the $k = \lfloor rN \rfloor$ **samples with the highest weights**, where r is the target ratio. Thus, the target ratio is enforced explicitly and exactly at every epoch. The continuous outputs are used as ranking scores, rather than through thresholding or weighted sampling. This design is also aligned with our reward formulation, prioritizing those with higher difficulty and uncertainty.
>
> As the revised manuscript can not be updated at the current stage, we will further clarify this in the final version.
> > - Sec. 3.1, lines 230: *"Samples with the top-k highest action weights are selected."*
> > - Alg. 1 in Appendix A, step 10: *"Select the k samples with the highest selection weights for next-epoch training."*
> - **Q4: Clarification of the MDP formulation.**
> - **A4:** We would like to clarify that the MDP formulation in Sec. 3.1 is a conceptual abstraction to provide a unified view of our method as a decision-making process. Its implementation is **explicitly instantiated by the subsequent modules** in our framework. Specifically, **(1) state space $S$ (lines 201-214)** corresponds to the current model status; **(2) action space $A$ (lines 216-219)** defines the selection weights; **(3) reward function (lines 173-185)** reflects the training objective; **(4) transition dynamics** are implicitly induced by model optimization; and **(5) initial state & horizon** denote the model initialization and the full training steps.
> - **Q5: Clarification on the PPO hyperparameters.**
> - **A5:** To ensure practicality and consistency, our implementation adopts a standard PPO configuration [a] across all experiments: $\gamma=0.99$, $\lambda=0.95$ for GAE, clipping $\epsilon=0.2$, and 10 PPO epochs per update. Importantly, we keep the same configuration across all experiments without task-specific tuning. We will add these details to Appendix D in the final version.
>
> [a] Proximal Policy Optimization Algorithm.
> - **Q6: Clarification on the non-stationarity of the MDP.**
> - **A6:** Insightful comment. We would like to clarify that the non-stationarity in our framework is motivated by the need to capture the evolving learning dynamics and data utility during training, enabling the agent to adapt its policy online as the target model learns. This design also avoids task-specific pretraining of separate agents or embedding models, making the framework more practical and plug-and-play.
>
> While this introduces non-stationarity, it is well-controlled and gradual: the target model is updated through incremental gradient steps, so the state and reward distributions shift smoothly over training. Under such co-evolution, PPO serves as a stable online optimizer, where clipped updates and advantage estimation help mitigate moderate drift. Empirically, this design yields stable training and consistent gains across diverse tasks. We will add this discussion to Appendix G.
> - **Q7: Clarification on Algorithm 1.**
> - **A7:** We will revise Alg. 1 accordingly to explicitly state Top-k subset construction in the final version.
> - **Q8: Clarification on the PPO's temporal machinery.**
> - **A8:** We would like to clarify that the sequential nature of our formulation is defined over the **training trajectory**, rather than explicit inter-sample interactions. While the agent assigns per-sample scores, its decisions affect the selected subsets, which in turn change the target model state and future rewards. Thus, the problem is inherently sequential over time, where PPO helps optimize long-horizon feedback during training.
>
> To further address the comment, we conducted additional experiments to compare with a learned scoring baseline that optimizes rewards via supervised regression (Table B-1). The results show our method consistently achieves better accuracy.
>
> **Table B-1: Comparison with the learned scoring approach on T-IN1k using R50.**
> ||30%|50%|70%|
> |-|-|-|-|
> |Learned Scoring|42.3|43.0|44.2|
> |Ours|**44.9**|**47.0**|**49.4**|
> - **Q9: Clarification on scoring strategy.**
> - **A9:** Thanks for the comments. We clarify that our method does **NOT** perform an extra full-dataset scoring pass each epoch. Instead, following common practice in dynamic data selection research (e.g., InfoBatch, UCB), the agent scores only the selected samples, so the efficiency claim remains valid.

---

> > ### Author Rebuttal · Reviewer_Pgoo · 2026-03-31
> >
> > Thank you for your responses. many of my questions are clarified, therefore I am increasing my score.

---

> > > ### Author Response · Authors · 2026-04-01
> > >
> > > Dear Reviewer Pgoo,
> > >
> > > We sincerely thank you for the thoughtful follow-up and constructive suggestions. We are glad that our clarifications have addressed your concerns. Thank you again for your time and effort in reviewing our work.
> > >
> > > Authors

---

### Official Review · Reviewer_MXqi · 2026-03-12

**Soundness:** 2
**Presentation:** 2
**Significance:** 2
**Originality:** 2
**Overall Recommendation:** 4
**Confidence:** 4

**Summary:**

This paper proposes Data Agent, a RL-based framework to select data on-the-fly. Specifically, at each training step, the agent generates rewards given the model state. The rewards are used to train the PPO-based agent to update the selection policy. The policy is co-evolved with the target model by using less data to enhance efficiency. Experiments show better results on vision and language tasks compared with previous methods.

**Compliance With Llm Reviewing Policy:**

Affirmed.

**Final Justification:**

I would like to maintian the score of 4 finally.

The paper partly addressed my concerns.

In the initial review, I wish to see more results on LLMs. The authors' rebuttal partly addressed my problems, but failed to conduct experiments on my aforementioned AoPS-instruct (600K) data for SFT, or ProRL for RL training. Therefore, I believe the practical empirical results seem not to be convincing enough to make me raise the score. However, I would like to maintain my score at 4 to support this paper because of the idea.

**Key Questions For Authors:**

See above weaknesses.

**Limitations:**

Yes.

**Strengths And Weaknesses:**

## Stengths
1. Using RL-based data selection method is a novel contribution to the online data selection community.
2. Online data selection is indeed crucial for efficiency given the evolving model states.

## Weaknesses
1. About the experiments: the main experiments are done in high-quality vision datasets, where most of the data should not be pruned, i.e., the data is already high-quality unlike real-scenario data like LLM pre-training data. If the selection method can be adapted to LLM pre-training, or at least post-training data selection, it would be more convincing. For example, the authors are suggested to conduct experiments on AoPS-instruct (600K) data for SFT, or ProRL for RL training
2. About the baseline methods: the compared methods are mostly in computer vision in small data regime. I wonder if the methods can be compared with other LLM pruning methods in post-training? Or, for vision tasks, can the Data agent be compared with data distillation methods?
3. About the efficiency. It seems that the online methods cause low efficiency due to training the PPO agent. When the data distribution is diverse, I wonder whether the method still works.
4. Related works: the authors are suggested to compare the method with other online data selection methods [1][2],etc.

[1] GREATS: Online Selection of High-Quality Data for LLM Training in Every Iteration.
[2] OPUS: Towards Efficient and Principled Data Selection in Large Language Model Pre-training in Every Iteration

---

> ### Author Rebuttal · Authors · 2026-03-31
>
> Dear Reviewer MXqi,
>
> We sincerely thank you for the insightful comments and constructive suggestions. We appreciate your recognition of our work's strengths, e.g, novelty, community contribution, and strong empirical results.
>
> We provide responses to address the comments as follows:
>
> - **Q1: Applicability to LLM training.**
> - **A1:** Thank you for pointing this out. We would like to clarify that while our method mainly focuses on vision tasks, we have extended it to LLM post-training settings in Sec. 4.4 (lines 363-376) using LLaMA-7B on MMLU and AlpacaEval 2.0. Results in Tab. 3 show consistent gains, validating the effectiveness of our approach in LLM post-training scenarios.
>
> Further extending our method to large-scale pre-training or RLHF pipelines requires additional system-level and algorithmic strategies, e.g., long-horizon credit assignment, which is beyond the scope of this work, but a promising direction for future work. Since the revised manuscript can not be updated at the current stage, we will include a discussion on these potential directions in Appendix G in the final version.
> - **Q2: Comparison with data distillation methods for vision tasks.**
> - **A2:** Thanks for the suggestion. We have conducted additional experiments to compare with representative data distillation methods. As shown in Table A-1, we utilized the reported results from prior works and followed the same training settings. The results show that our method consistently outperforms these approaches.
>
> Notably, Table A-2 shows a significant efficiency advantage. This is because data distillation typically involves expensive synthetic data generation and trajectory matching. In contrast, our method directly learns a lightweight policy online, achieving high efficiency. Moreover, most existing distillation methods are specifically designed for classification tasks, while our method  is task-agnostic and generalizes across diverse tasks, demonstrating broader applicability.
>
> **Table A-1:** Comparison with data distillation methods using ConvNet with a 20% selection ratio. - means results are lower than the random baseline.
> ||C-10|C-100|
> |-|-|-|
> |Random|78.4|42.8|
> |TESLA [a]|-|49.2|
> |MTT [b]|-|49.0|
> |FTD [c]|-|49.7|
> |DATM [d]|85.5|57.5|
> |Ours|**87.3**|**60.1**|
>
> **Table A-2:** Efficiency comparison on CIFAR-10 with a 20% selection ratio.
> ||Cost (h)|
> |-|-|
> |TESLA [a]|>9.6|
> |MTT [b]|30.0|
> |FTD [c]|>17.5|
> |DATM [d]|9.6|
> |Ours|**0.02**|
>
> Reference:
>
> [a] Cui, J., et al. "Scaling up dataset distillation to imagenet-1k with constant memory."
>
> [b] Cazenavette, G., et al. "Dataset distillation by matching training trajectories."
>
> [c] Du, J., et al. "Minimizing the accumulated trajectory error to improve dataset distillation."
>
> [d] Guo, Z., et al. "Towards lossless dataset distillation via difficulty-aligned trajectory matching."
> - **Q3: Clarification on the efficiency and robustness under diverse data distributions.**
> - **A3:** Thanks for the question. We would like to clarify that, while our method introduces an online agent, the additional computational overhead is minimal (less than ~3% as shown in Table 2/9). This is because the agent is extremely lightweight, and the reward signal is obtained from standard forward passes. Thus, our method does **NOT** require any expensive procedures.
>
> Regarding robustness under diverse data distributions, we conducted additional experiments on the long-tailed dataset Places-LT. As shown in Table A-3, our method maintains lossless performance with reduced training data volume.
>
> **Table A-3:** Evaluation on long-tail distribution using Places-LT with the closed-set setting. We report overall accuracy (%).
> ||80%|90%|100%|
> |-|-|-|-|
> |Ours|34.6|36.1|34.5|
> - **Q4: More suggested references in Related Work.**
> - **A4:** Thank you for suggesting more related works.  Since the revised manuscript can not be updated at the current stage, we will include these references in Section 2.2 in the final version. Specifically,
>   - Sec 2.2, paragraph 1, add references *"Differently, OPUS (Wang et al., 2026) proposes an optimizer-induced dynamic selection, which formulates data utility through optimizer-induced update dynamics, and GREATS (Wang et al., 2024) optimizes batch quality via Taylor expansion to reduce validation loss."*

---

> > ### Author Rebuttal · Reviewer_MXqi · 2026-04-01
> >
> > In the initial review, I wish to see more results on LLMs. The authors' rebuttal partly addressed my problems, but failed to conduct experiments on my aforementioned AoPS-instruct (600K) data for SFT, or ProRL for RL training. Therefore, I believe the practical empirical results seem not to be convincing enough to make me raise the score. However, I would like to maintain my score at 4 to support this paper because of the idea.

---

> > > ### Author Response · Authors · 2026-04-02
> > >
> > > Dear Reviewer MXqi,
> > >
> > > We sincerely thank Reviewer MXqi for the **thoughtful follow-up and for maintaining support for the acceptance of our paper.** We also appreciate your suggestion on strengthening the LLM empirical evaluation of our work.
> > >
> > > Due to the limited rebuttal timeline and the cost of large-scale SFT experiments, we were unable to include these results in the initial rebuttal. Following your suggestion, we now provide the completed results on AoPS-instruct (600K) for SFT.
> > >
> > > - **Q5: Additional results on AoPS-instruct (600K).**
> > > - **A5:** We conducted SFT on Qwen2.5-1.5B with LoRA on AoPS-Instruct (600k), and evaluated on the challenging LiveAoPSBench (Dec-2024) set using zero-shot CoT prompting. As shown in Table A-4, despite using only 70-80% of the training data, our method achieves competitive or slightly improved performance compared with full-data training. This suggests that our approach can effectively identify informative samples even in high-quality LLM SFT settings.
> > >
> > >
> > > **Table A-4:** Evaluation on AoPS-instruct (600K) using Qwen2.5-1.5B (accuracy, %).
> > > |Selection Ratio (%)|70|80|100|
> > > |-|-|-|-|
> > > |Ours|8.5%|8.7%|7.7%|
> > >
> > > We will include these results in the final version in Appendix, and further discuss extensions to broader LLM settings (e.g., RL-based training such as ProRL) in Appendix G.

---

### Decision · Program_Chairs · 2026-04-30

**Decision:**

Accept (regular)

**Comment:**

This paper proposes Data Agent, a reinforcement-learning-based framework for dynamic data selection during training. The core idea is to formulate data selection as a sequential decision-making problem, where a lightweight PPO agent learns to score samples based on loss difficulty and confidence uncertainty, with an adaptive weighting mechanism that shifts emphasis between signals as training progresses. The authors evaluate the approach across image classification, object detection, semantic segmentation, and LLM instruction tuning, reporting consistent improvements in training efficiency while preserving or exceeding full-data performance.

The rebuttal phase was notably productive. The authors added over 20 new experimental results across eight supplementary tables, addressing concerns about baseline coverage, methodological clarity, robustness under extreme settings, and cross-architecture generalization. Two reviewers who initially expressed reservations (R2, score 3) and (R1, score 4) both acknowledged substantive resolution of their primary concerns. R3 and R4, who scored 5 (Accept), confirmed full resolution. The paper's contribution — an end-to-end RL formulation for co-evolving data selection with model training — is novel within the data pruning literature and the empirical validation across diverse task modalities is unusually broad.